# Cnot4 heterozygosity attenuates high fat diet-induced obesity in mice and impairs PPARγ-mediated adipocyte differentiation

Tomokazu Yamaguchi[1], Midori Hoshizaki[1], Yumiko Imai[2], Tadashi Yamamoto[3], Keiji Kuba[1]*

1 Department of Pharmacology, Kyushu University Graduate School of Medical Sciences, Fukuoka, Japan, 2 Department of Medical Infection System, Research Institute Nozaki Tokushukai Hospital, Osaka, Japan, 3 Cell Signal Unit, Okinawa Institute of Science and Technology, Okinawa, Japan

* kuba.keiji.815@m.kyushu-u.ac.jp

## Abstract

Adipocyte differentiation is crucial for formation and expansion of white adipose tissue and is also associated with the pathologies of obesity. CNOT4 is an E3 ubiquitin ligase and also contains RNA binding domain. In mammals CNOT4 has been suggested to interact with CCR4-NOT complex, a major executor of mRNA poly(A) shortening. While several subunits within the CCR4-NOT complex were shown to be involved in obesity and energy metabolism, the roles of CNOT4 in obesity remain unexplored. In this study, we generated and analyzed Cnot4 knockout mice and found that Cnot4 heterozygous (Cnot4 Het) mice exhibit resistance to high fat diet-induced obesity, including significant reduction in adipose tissue mass and hepatic lipid depots. However, Cnot4 Het did not affect mRNA expression of metabolic genes as well as serum lipid levels or glucose tolerance. On the other hand, Cnot4 Het fibroblasts significantly reduced the capability of differentiation into adipocytes and down-regulated adipogenic gene expression compared to wild type fibroblasts. Mechanistically, heterozygous deletion of Cnot4 down-regulated the transcriptional activity through decreased binding of PPARγ to promoter region of the target gene, thereby suppressing up-regulation of adipocyte marker gene expression in response to rosiglitazone, a PPARγ agonist. These results suggest that CNOT4 mediates adipocyte differentiation during formation and growth of adipose tissue partly through positively regulating transcriptional activity of PPARγ.

## Introduction

Obesity, a major risk factor for chronic diseases including type 2 diabetes, dyslipidemia, and cardiovascular conditions, is characterized by adipose tissue enlargement. The increase of adipose tissue mass occurs through two mechanisms; hypertrophy (increase in adipocyte size) and hyperplasia (differentiation from preadipocytes) [1,2]. Hypertrophy of adipocytes induces mechanical stress to neighboring cells and

**Data availability statement:** All relevant data are within the paper and its Supporting Information files.

**Funding:** K.K. is supported by the Japan Society for the Promotion of Science [24K02215], and Tomokazu Yamaguchi is supported by the Japan Society for the Promotion of Science [23K06146].

**Competing interests:** The authors have declared that no competing interests exist.

insufficient vascularization/hypoxia and promotes chronic inflammation and fibrosis, leading to unhealthy obesity, such as exacerbation of insulin insensitivity and dyslipidemia [3–5]. In contrast, hyperplasia increases the number of adipocytes keeping them in relatively smaller size and maintains local vascularization and production of adipokines including adiponectin and leptin, which confers insulin sensitive and lipolytic properties [1,6,7]. The nuclear receptor PPARγ plays a pivotal role in hyperplasia by promoting differentiation of preadipocytes into adipocytes [8,9]. Activated by ligands such as polyunsaturated fatty acids and prostaglandins, PPARγ forms a heterodimer with the retinoid X receptor. This complex binds to peroxisome proliferator response elements in gene promoters, driving the expression of adipogenic genes such as AP2, GLUT4, LPL and adiponectin, thereby promoting adipogenesis [9–12].

CNOT4 is an E3 ubiquitin ligase having a RING domain for interacting with E2 ubiquitin-conjugating enzymes and ubiquitinates target proteins [13], and CNOT4 also has an RNA-recognition motif and a Caf40 (CNOT9)-binding motif, which mediates its binding to the carbon catabolite repression 4-negative on TATA-less (CCR4-NOT) complex [14,15]. CCR4-NOT complex is a large multimeric protein complex, which catalyzes shortening of poly(A) tail for mRNA degradation and also regulates transcription and translation [16]. Thus, CNOT4 has also been shown to contribute to various process of gene expression, such as translational quality control in ribosome [17,18] and transcriptional elongation or epigenetic regulation in the nucleus [19–22]. On the other hand, mammalian CNOT4 is a temporal but not stable interactor of CCR4-NOT complex [23]. The physiological relevance of interaction between CNOT4 and CCR4-NOT complex remains elusive, whereas CNOT4 was recently shown to be involved in poly(A) shortening during meiosis [24]. Previous reports on gene knockout mice of CCR4-NOT components, such as CNOT3, CNOT6L and CNOT7, have demonstrated that CCR4-NOT complex plays roles in suppressing diet-induced obesity and improving metabolic disorder through degradation of target mRNAs [25–27]. Thus, while the significance of CCR4-NOT complex in regulating metabolic genes in liver and adipose tissues has been proposed, the roles of CNOT4 in obesity are elusive.

In this study, we investigated whether Cnot4 regulates high fat diet (HFD)-induced obesity and metabolic disorder. We generated Cnot4 knockout mice by gene targeting and confirmed that Cnot4 null embryos are lethal. Cnot4 heterozygous knockout (Cnot4 Het) mice are viable with slightly reduced body size and weight. Cnot4 Het mice exhibited resistance to HFD-induced obesity. However, elevated serum lipid levels and glucose intolerance were not improved in Cnot4 Het mice. Heterozygous deletion of Cnot4 in mouse embryonic fibroblasts (MEFs) suppressed adipocyte differentiation and down-regulated adipogenic gene expression partly through reduced transcriptional activity of PPARγ. Thus, CNOT4 positively regulates transcriptional activity of PPARγ during adipocyte differentiation in obesity.

## Materials and methods

### Heterozygous knockout of Cnot4 in mice and animal experiments

For gene targeting of Cnot4 in mice, a targeting vector was constructed to replace exons 2 of the murine *Cnot4* gene to flanking exon 2 with two loxP sites and inserting

PGK-Neo cassette. The linearized construct was electroporated into A9 embryonic stem (ES) cells, obtained targeted ES clones and generated chimeric mice, as described previously [28]. The targeted allele was confirmed by PCR, and F1 mice were backcrossed for 5 times onto a C57BL/6 background. The mice carrying heterozygous floxed allele of Cnot4 (*Cnot4*<sup></sup>) — see below — were crossed with *CAG-Cre*<sup></sup> mice to generate Cnot4 Het mice. Mice were genotyped by PCR and maintained at the animal facilities of Akita University and Kyushu University. Sequences of the forward and reverse primers for genotyping PCR are shown in S1 Table. Mice were fed with a normal diet (CA-1, CLEA) or a high fat diet (D12492, EPS Holding Inc.). Throughout the high-fat diet intervention, animals were monitored daily for signs of distress. If a mouse exhibited severe weakness, reduced mobility, or persistent abnormal behavior, euthanasia was performed to prevent unnecessary suffering. Mice were euthanized via i.p. injection of an overdose of anesthesia containing ketamine (200 mg kg$^{-1}$) and xylazine (20 mg kg$^{-1}$) prior to tissue collection. All animal experiments conformed to the Guide for the Care and Use of Laboratory Animals, Eighth Edition, updated by the US National Research Council Committee in 2011, and approvals of the experiments were granted by the ethics review board of Akita University and Kyushu University.

## Mouse embryo analysis

To analyze mouse embryo, embryos at 10.5–14.5 days post coitus (dpc) were harvested from the uteri of pregnant mice of heterozygous x heterozygous breeding. Genomic DNA was extracted from tail of each embryo or yolk sac encompassing each embryo and genotyped by PCR. For morphological observation, embryos were fixed with 4% formalin and the high-resolution images of embryos were taken with microscope (Keyence).

## Blood analysis

For glucose tolerance tests, mice were fasted for 16 h. Glucose (0.5 mg/g body weight) was intraperitoneally injected, and blood glucose was measured from tail blood using the Glucose Pilot Meter (Technicon International Inc.). Serum levels of triglyceride and cholesterol were measured using FUJI DRI-CHEM slide kits (Fujifilm corporation).

## Histology

To evaluate histology of liver, white adipose tissue, and brown adipose tissue, mice were euthanized with overdose of anesthesia, and tissues were excised. Tissues were fixed with 4% formalin and embedded in paraffin. Five-µm-thick sections were prepared and stained with Hematoxylin & Eosin (H&E). The high-resolution images (x200 magnification) of the tissue sections stained with H&E were taken with microscope (Nikon). For quantification of adipocytes in WATs, three randomly chosen fields of each section were taken with microscope (Keyence), and the size of adipocytes in field was measured by Hybrid Cell Count measurement software (Keyence). The average of size range percentages on three fields of each section were used as the one individual value.

## Cell cultures

For adipocyte differentiation, MEFs were isolated from 13.5 dpc embryos from the crossing of Cnot4 Het mice with wild type (WT) mice, and the genotypes were determined by PCR. Cells were cultured in 10% FBS containing Dulbecco's modified Eagle medium (DMEM), and cell growth was evaluated by counting cell number at the indicated time points after plating 10,000 cells in 24 well plate. To differentiate into adipocytes, 2-day post-confluent MEFs were treated with differentiation medium (DMEM with 10% FBS supplemented with 10 µg/ml Insulin, 0.5 mM 3-Isobutyl 1-methylxanthine (IBMX), 1 µM dexamethasone, and 10 µM rosiglitazone) for 2 days, then for 6 days in medium supplemented with insulin and rosiglitazone. Cells were harvested for RNA analysis and lipid staining on days 5 and 8 after the initiation of differentiation medium treatment. For lipid staining, the cells were fixed with 10% formalin for 30 min at 37°C and stained with Oil Red O. For quantification of stained area, three randomly chosen fields of each cell were taken with microscope (Keyence), and area stained with Oil Red O in field was measured by ImageJ software. The average area size of three fields were

used as the one individual value. The lipid content was quantified by measuring the absorbance at 490 nm of Oil Red O extracted with isopropanol. Transcriptional activity of PPARγ was analyzed in immortalized MEFs. WT and Cnot4 Het MEFs were immortalized by infection with lentivirus made from pLenti CMV/TO SV40 small + large T vector (plasmid no. 22298; Addgene). Rosiglitazone treatment was done at 10 μM for 6 hours.

## RNA analysis

Mouse liver and WATs were homogenized in TRIzol reagent (Invitrogen), and total RNA was extracted by acid guanidinium thiocyanate-phenol–chloroform method. Total RNA of MEFs were purified by Direct-zol RNA Microprep (Zymo Research). cDNA was synthesized using the PrimeScript RT reagent kit (RR037; TAKARA) and quantitative real-time PCR was run in 96 well plates using a SYBR Premix ExTaq II (RR820; TAKARA) according to the instructions of the manufacturer. Relative gene expression levels were quantified by using the Thermal Cycler Dice Real Time System II software (TAKARA). Sequences of the forward and reverse primers of the genes studied are shown in S2 Table.

## Western blotting

MEF proteins were extracted with TNE lysis buffer (50 mM Tris, 150 mM NaCl, 1 mM EDTA, 1% NP40, protease inhibitor (Complete Mini; Roche), 20 mM NaF, 2 mM $Na_3VO_4$). After sonication and denaturation with LDS sample buffer (Invitrogen) at 70°C, proteins were electrophoresed on NuPAGE bis-tris precast gels (Invitrogen) and transferred to nitrocellulose membranes (0.45 μm pore; Invitrogen). Total of loading protein ware visualized by staining of protein-blotted membrane with ponceau solution (Sigma). Membranes were probed with anti-PPARγ antibody (CST, 81B8, 1:1000 diluted), anti-CNOT4 antibody (Sigma, HPA005737, 1:1000 diluted), GAPDH (CST, 14C10, 1:3000 diluted), and H3 (CST, D1H2, 1:1000 diluted). The blotting bands visualized with ECL reagent (Bio-Rad) using ChemiDoc Touch Imaging System (Bio-Rad). Subcellular fractionation and protein extraction from MEFs were performed by NE-PER Nuclear and Cytoplasmic Extraction Kit (Thermo Fisher Scientific) according to the instructions of the manufacturer.

## Luciferase assay

MEFs were seeded in 24-well plates ($3.0 \times 10^4$ cells/well), and cells were transfected with 500 ng of reporter plasmid DNA containing PPARγ response element following luciferase sequence (plasmid no. 1015; Addgene) using Lipofectamine 3000 (Invitrogen) at 24 hours after plating. Twenty-four hours after plasmid transfection, MEFs were treated with 1 μM of rosiglitazone for 6 hours. MEFs were washed with PBS once, lysed with lysis reagent included the Dual-luciferase assay system kit (Promega) and then luciferase assay was performed following the instructions of the manufacturer. Luciferase activity was measured using GloMAX-Multi Detection System (Promega). The luciferase activity was normalized by total protein concentration quantified by BCA Protein Assay Kits (Thermo Fisher Scientific).

## ChIP assay

ChIP assays were performed using the ChIP-IT High Sensitivity (Active Motif). WT and Cnot4 Het MEFs were crosslinked with Fixation solution containing 1.1% formaldehyde for 15 min, quenched with Stop solution, lysates harvested, and chromatin DNAs were shared to 300–600 bp size by using the Bioruptor (COSMO BIO). 0.25 μg of Anti-PPARγ antibodies (CST, 81B8) was used to immunoprecipitate the DNA/protein complex. IgG (Active Motif) was used as a negative control. Crosslink reversed samples were treated with Proteinase K and the DNA purified and analyzed by qPCR. The qPCR primers were designed in the region of aP2 genomic locus [29].

## Statistical analyses

Data are presented as mean values ± SEM. Statistical significance between two experimental groups was determined using Student's two-tailed t-test. Comparisons of parameters among groups were analyzed by one-way analysis of

variance (ANOVA), followed by Sidak's multiple-comparisons test. When a comparison is done for groups with two factor levels, two-way ANOVA with Sidak's multiple-comparisons test were used. P < 0.05 was considered significant. Statistical analyses were performed using GraphPad Prism version 10.4.1 (GraphPad Software).

## Results

### Cnot4 homozygous knockout mice are embryonic lethal

To determine the physiological roles of CNOT4 *in vivo*, we generated Cnot4 knockout mice. Murine *Cnot4* gene was disrupted in embryonic stem cells by homologous recombination with a targeting vector flanking exon 2 with two loxP sites and inserting PGK-Neo cassette, and Cnot4*flox-Neo* mice crossed with Cre deleter mice (*CAG-Cre*Tg/+) to obtain Cnot4 heterozygous mice (*Cnot4*+/- or Cnot4 Het) (Fig 1A). We confirmed deletion of Exon 2 of *Cnot4* gene in the embryos from crossing of Cnot4 Het mice by genomic PCR (Fig 1B). Cnot4 Het mice are viable with normal fertility, whereas *Cnot4*-/- (Cnot4 KO) newborn pups were not obtained. Analysis of embryonic staging revealed that Cnot4 KO embryos are lethal at 10.5 days post coitus (dpc) (Fig 1C and 1D). Thus, Cnot4 is essential for early embryonic development.

### Cnot4 haploinsufficiency confers resistance to high fat diet-induced obesity without affecting lipid or glucose metabolism

Since heterozygous knockout of Cnot3 decreases body size and confers resistance to high fat diet (HFD)-induced obesity [25,30], we anticipate that Cnot4 heterozygous mice are also resistant to obesity. We confirmed that expression of Cnot4 mRNA was decreased in the livers of Cnot4 Het mice approximately to the half of that in Cnot4 WT mice (Fig 2A). At 20 weeks of age under normal diet (ND)-feeding condition, Cnot4 Het mice exhibited a slight reduction in the body size (Fig 2B); the nose to anus length (N-A length) of Cnot4 Het mice was reduced by approximately 5% compared with WT mice and Cnot4 Het mice weighed ~13% less than WT mice (Fig 2C and 2D). To address whether Cnot4 heterozygous deletion affect the development and postnatal growth of organs, we measured the weight of various organs in Cnot4 Het mice and normalized with N-A length. The organ weight of heart, skeletal muscle, liver, epididymal white adipose tissue (WATs) were not altered in Cnot4 Het mice (Fig 2E–2G), whereas the weight per N-A length of brown adipose tissue was slightly decreased in Cnot4 Het mice compared with WT mice (Fig 2H).

We next challenged Cnot4 Het mice with a high-fat diet (HFD). After 12 weeks of HFD feeding, Cnot4 Het mice showed significant reduction in body size and weight compared with WT mice (Fig 2B–2D). Importantly, the weight gain of liver, WATs and BATs were significantly suppressed in Cnot4 Het mice compared with WT mice after HFD feeding (Fig 2F–2H). Consistently, histological analyses showed that fat depots in the liver, WATs, and BATs were reduced in Cnot4 Het mice compared with WT mice (Fig 2I). The size of adipocytes in WATs was diminished in Cnot4 Het mice compared with WT mice (Fig 2J). We further examined if any alteration in lipid or glucose metabolism. When we measured mRNA expression of the genes associated with lipid and cholesterol metabolism, there were no significant changes in expression of those genes in WATs and livers (Fig 3A and 3B). In addition, despite resistance to obesity in Cnot4 Het mice, serum levels of triglyceride and total cholesterol in Cnot4 Het mice were comparable to those in WT mice (Fig 3C), and glucose tolerance test indicated that glucose tolerance was not improved in Cnot4 Het mice (Fig 3D). Thus, Cnot4 haploinsufficiency mitigates HFD-induced obesity but does not improve dyslipidemia and glucose intolerance. We further investigated whether Cnot4 regulates the expression of target mRNAs of Cnot3 and Cnot6l in previous reports [25,26]. Previous studies have shown that mRNAs encoding the genes of energy metabolism (*Pdk4*, *Pgc1a*, *Ucp1*, and *Igfbp1*) and hepatokine (*Gdf15* and *Fgf21*) were significantly up-regulated in *Cnot3*+/- hepatocytes and Cnot6l-deleted hepatocytes, respectively [25,26]. Heterozygous deletion of *Cnot4* up-regulated expression of *Pgc1a* and *Igfbp1* in the livers (Fig 3E), whereas there were no changes in expression of other target genes. Thus, the mechanism of Cnot4-mediated gene regulation is likely distinct from CCR4-NOT-mediated deadenylation and subsequent RNA decay. In addition, altered mRNA expression of *Pgc1a*

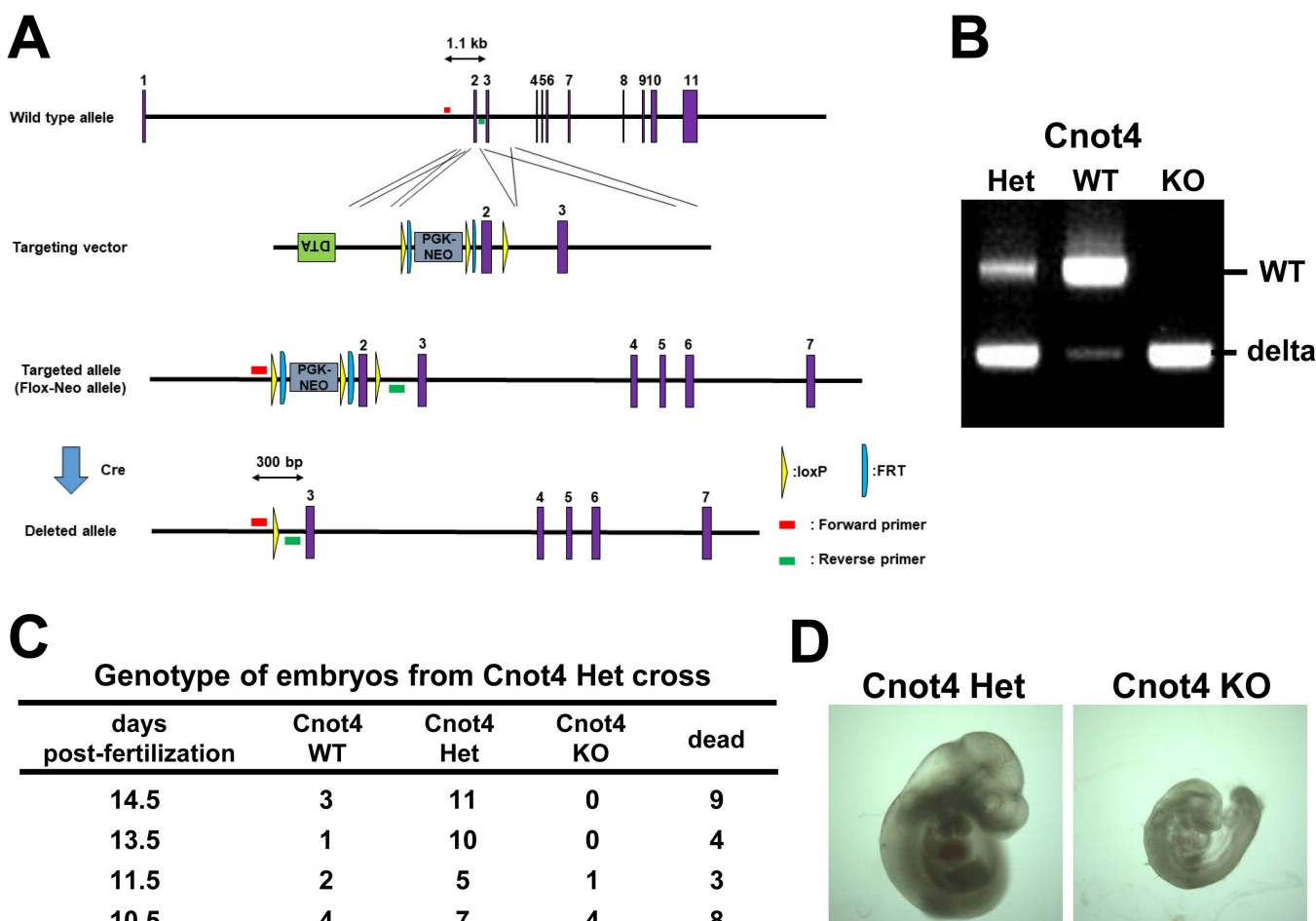

**Fig 1. Cnot4 homozygous knockout mice are embryonic lethal.** (A) Strategy for Cnot4 gene knockout in mice. Exon 2 of the murine *Cnot4* gene was flanked with two loxP sites and inserted PGK-Neo cassette by homologous recombination in ES cells. Cnot4 Flox-Neo mice were crossed with *CAG-Cre*[Tg/+] mice to delete exon 2. (B) Genotyping PCR results of embryos at 10.5 days post coitus (dpc) in the uterus of *Cnot4*[+/-] female mice crossed with *Cnot4*[+/-] male mice. The DNA band patterns showing wild type (WT), Cnot4 heterozygous deletion (Het), and Cnot4 homozygous deletion (KO) are presented. (C) The ratio of genotypes in embryos at various stages. Embryos were harvested from uterus of *Cnot4*[+/-] female mice crossed with *Cnot4*[+/-] male mice. (D) Representative photograph of mouse embryos at 10.5 dpc. Bars indicate 500 μm.

and *Igfbp1* may be potentially related to the phenotypes of resistance to obesity but no obvious changes in dyslipidemia and glucose intolerance in Cnot4 Het mice.

### Cnot4 mediates differentiation of mouse embryonic fibroblasts to adipocytes

Since our RNA-seq analysis of WATs showed that expression of the genes related to various developmental process was down-regulated in Cnot4 Het WATs under ND condition (not shown), we reasoned that heterozygous deletion of Cnot4 partly impaired adipocyte differentiation thereby leading to decreased fat depots in adult mice without any improvement of lipid or glucose metabolism. We thus examined whether Cnot4 heterozygosity influences differentiation of preadipocytes into mature adipocytes. To this end, we exploited an experimental system to derivate adipocytes from mouse embryonic fibroblasts (MEFs) [31]. We isolated the MEFs from Cnot4 Het embryos at 13.5 dpc and confirmed that Cnot4 expression levels were reduced in Cnot4 Het MEFs by half of those in WT MEFs (Fig 4A). Cnot4 Het MEFs did not show apparent

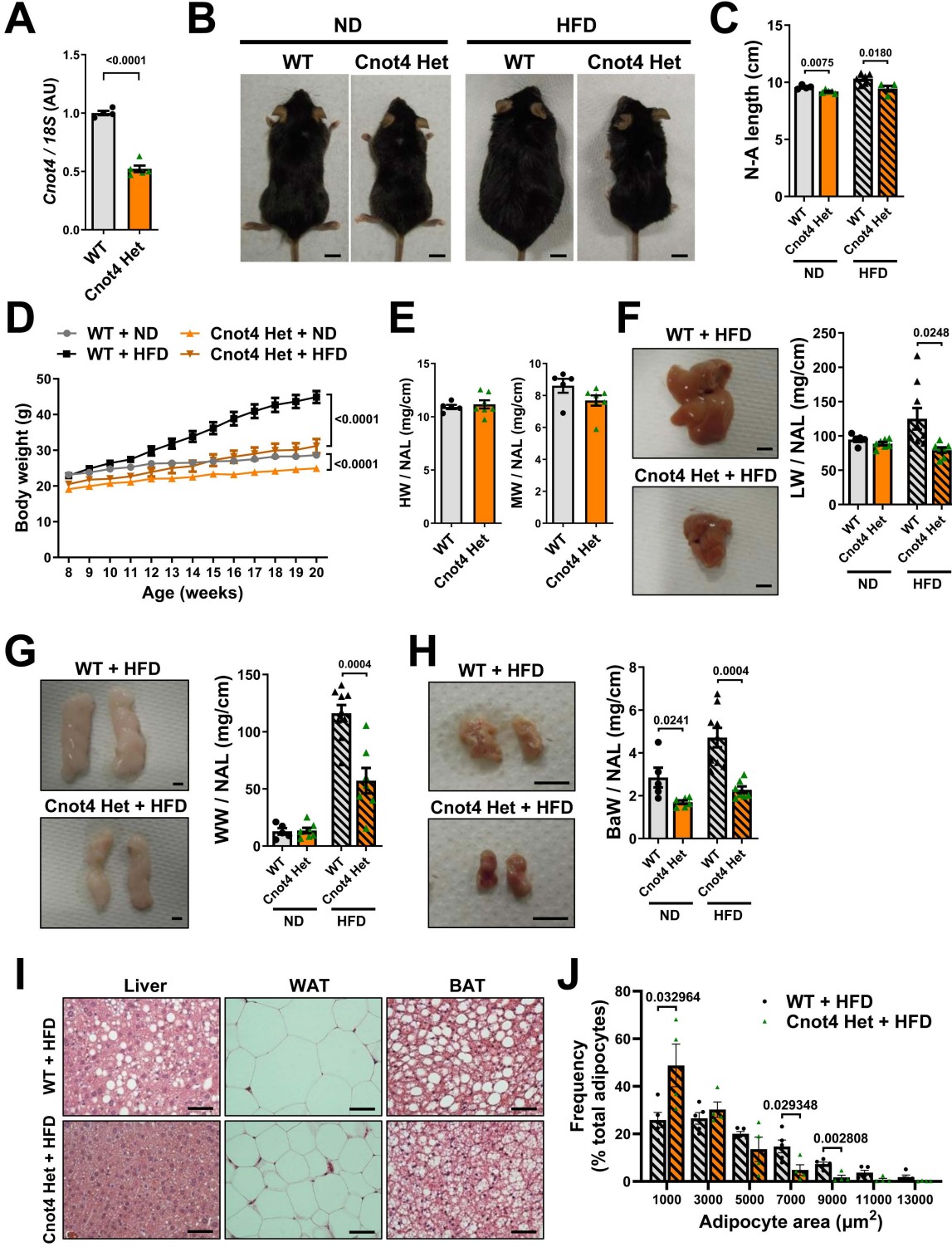

**Fig 2. Cnot4 heterozygosity confers resistance to high fat diet-induced obesity.** (A) mRNA expression of Cnot4 in mouse livers. WT (n = 4), Cnot4 Het (n = 5). (B) Representative photographs of mice. Bars indicate 1 cm. (C) Nose-to-anus (N-A) length of 20-week-old mice. WT + ND (n = 4), Cnot4 Het + ND (n = 5), WT + HFD (n = 6), Cnot4 Het + HFD (n = 4). (D) Body weight curve of mice from 8 to 20 weeks after birth. WT + ND (n = 4), Cnot4 Het + ND (n = 6), WT + HFD (n = 10), Cnot4 Het + HFD (n = 10). (E) Weight of hearts (HW) and tibialis anterior muscles (MW) normalized with N-A length. WT (n = 5),

Cnot4 Het (n = 7). (F-H) Representative photographs (left) and weight (right) of mouse livers (F), white adipose tissues (WATs) (G), and brown adipose tissues (BATs) (H). Bars indicate 5 mm. WT + ND (n = 5), Cnot4 Het + ND (n = 7), WT + HFD (n = 9), Cnot4 Het + HFD (n = 7). (I) Representative histological images of livers, WATs and BATs. Bars indicate 50 μm. All values are means ± SEM. (J) Adipocyte size quantification in WATs. WT + HFD (n = 5), Cnot4 Het + HFD (n = 4). Two-tailed unpaired *t*-test (A, C, E-H, and J) and Two-way ANOVA with Sidak's multiple-comparisons test (D) were used to detect significance. Numbers above or on the right of square brackets show significant *P*-values.

changes in cell proliferation compared with WT MEFs, albeit non-significant reduction in cell number was observed in Cnot4 Het MEFs (Fig 4B). When MEFs were induced to differentiate into adipocytes with adipogenic factors including insulin, phosphodiesterase inhibitor (3-Isobutyl-1-methylxanthine (IBMX)), glucocorticoid (dexamethasone), and PPARγ agonist (rosiglitazone), Cnot4 Het MEFs exhibited significant reduction in accumulation of lipid compared to WT MEFs (Fig 4C and 4D). Consistently, expression of adipocyte-specific genes in Cnot4 Het MEFs were lower than those of WT MEFs after adipocyte differentiation (Fig 4E). Intriguingly, mRNA expression of fatty acid synthesis genes, including *Srebf1c*, *Fasn* and *Acaca*, were significantly down-regulated in Cnot4 Het MEFs (Fig 4F), whereas there were no changes in expression of those genes in the WATs of Cnot4 Het mice with HFD feeding (Fig 3B). On the other hand, expression of cholesterol biosynthesis-related genes (*Hmgcs1* and *Mvk*) were not altered in adipocyte-differentiating Cnot4 Het MEFs (Fig 4F), which is the same as the WATs of Cnot4 Het mice (Fig 3B). Thus, Cnot4 mediates differentiation of embryonic fibroblasts to adipocytes likely through up-regulating mRNA expression of the adipogenic genes.

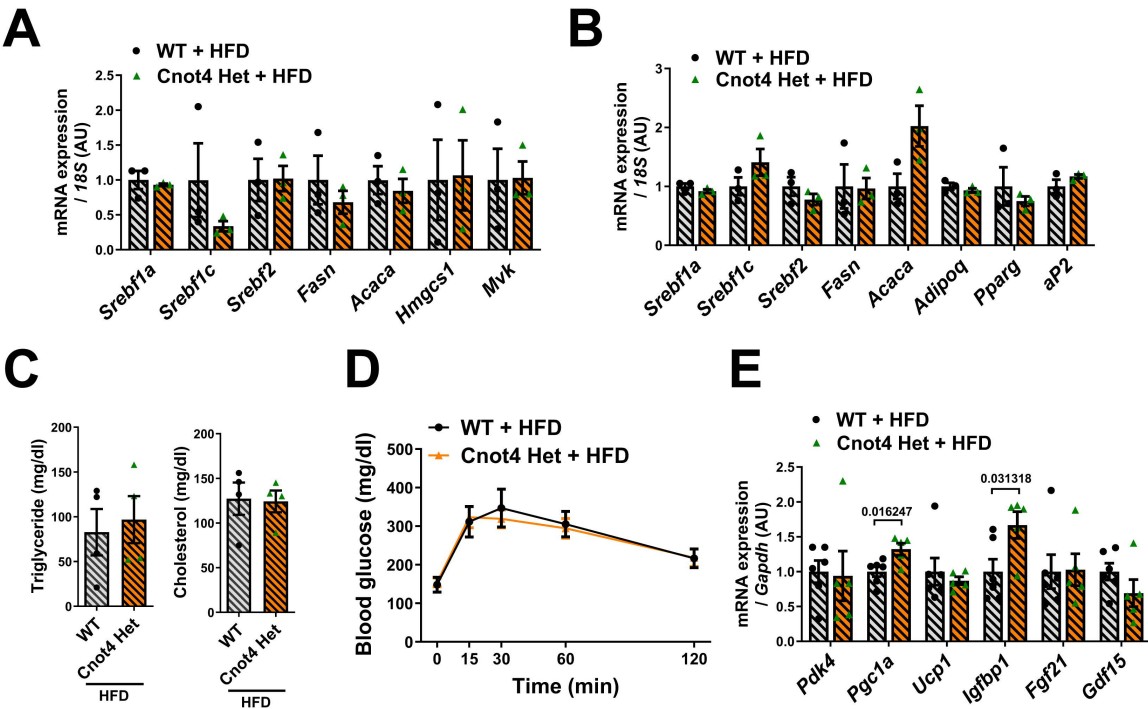

**Fig 3. Cnot4 Het does not affect mRNA expression of metabolic genes as well as serum lipid levels or glucose tolerance.** (A, B) mRNA levels of lipid metabolic genes in mouse livers (A) and WATs (B). (n = 3). (C) Blood triglyceride and cholesterol level in mice. (D) Glucose tolerance test. Mice were fasted for 16 hours prior to the experiment. Blood glucose levels in mice were measured at the indicated times following intraperitoneal glucose injection. WT + HFD (n = 7), Cnot4 Het + HFD (n = 11). (E) Levels of mRNAs encoding the genes related to energy metabolism and hepatokine in mouse livers. WT + HFD (n = 5), Cnot4 Het + HFD (n = 6). All values are means ± SEM. Two-tailed unpaired *t*-test (A-C, E) and Two-way ANOVA with Sidak's multiple-comparisons test (D) were used to detect significance. Numbers above of square brackets show significant *P*-values.

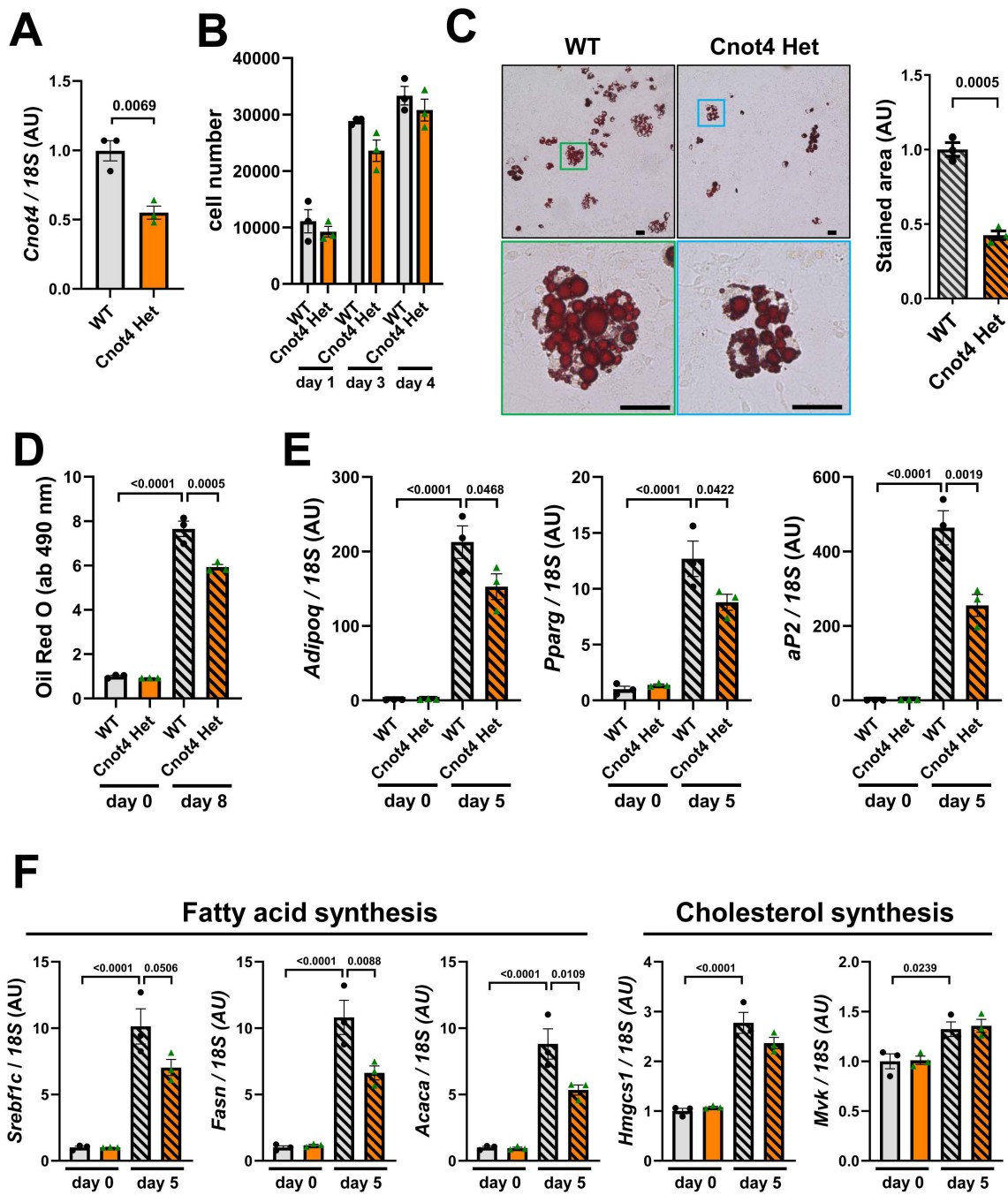

**Fig 4. Cnot4 mediates differentiation into adipocytes from mouse embryonic fibroblasts.** (A) mRNA expression of Cnot4 in mouse embryonic fibroblasts (MEFs) (n = 3). (B) Cell growth of MEFs. After seeding 10,000 cells, the cells were counted at the indicated time points (n = 3). (C) Representative images of adipocytes stained with Oil Red O (left). The cells were stained at 8 days after induction of differentiation. Higher magnification of the colored rectangles (top) is shown in the bottom panel. Bars indicate 100 μm. The areas of Oil Red O staining were measured (right) (n = 3). (D) Quantification of lipid abundance in adipocytes. Oil Red O was eluted from the stained MEF-derived adipocytes, and the absorbance at 490 nm was measured (n = 3). (E) mRNA levels of adipocyte marker genes. MEFs-derived adipocytes at 5 days after differentiation (n = 3). (F) mRNA expression of the genes of fatty acid and cholesterol synthesis. All values are means ± SEM. Two-tailed unpaired $t$-test (A-C). One-way ANOVA with Sidak's multiple-comparisons test (D-F). Numbers above square brackets show significant $P$-values.

## Cnot4 haploinsufficiency impairs rosiglitazone-induced transcriptional activity of PPARγ

To gain mechanistic insights into Cnot4-mediated adipocyte differentiation, we investigated whether Cnot4 haploinsufficiency affects the transcriptional activity of PPARγ, which is a master regulator of adipocyte differentiation [8,9]. We examined the protein expression of PPARγ in MEFs, and observed that Cnot4 Het MEFs express PPARγ protein (Fig 5A). To address transcriptional activity of PPARγ, we treated MEFs with rosiglitazone, a known PPARγ agonist [32]. While rosiglitazone up-regulated the mRNA levels of *Adipoq* and *Pparg* in WT MEFs, Cnot4 Het MEFs showed decreased expression of *Adipoq* and *Pparg* genes (Fig 5B). Although rosiglitazone did not up-regulate *aP2* gene expression in WT MEFs, *aP2* expression was significantly suppressed in Cnot4 Het MEFs under basal conditions (Fig 5B). We next conducted promoter assay using reporter plasmid harboring the genomic region of PPARγ response element (PPRE) following luciferase sequence, and it revealed that transcriptional activity of PPARγ was markedly down-regulated in Cnot4 Het MEFs compared to WT MEFs under basal condition (Fig 5C). Rosiglitazone slightly but significantly increased PPRE promoter activity in WT MEFs. By contrast, Cnot4 Het MEFs impaired rosiglitazone-induced elevation of PPRE promoter activity (Fig 5C). To determine whether heterozygosity of Cnot4 affect the nuclear translocation of PPARγ, subcellular fractionation was conducted. The results showed that the amount of PPARγ protein was not decreased but rather increased in the nuclear fractions of Cnot4 Het MEFs compared with those of WT MEFs (Fig 5D). In the meantime, chromatin immunoprecipitation (ChIP) assay showed that the binding of PPARγ protein to *aP2* gene locus was reduced in Cnot4 Het MEFs compared with WT MEFs (Fig 5E). Therefore, Cnot4 positively regulates basal and agonist-induced transcriptional activity of PPARγ by recruiting it to target gene locus, which likely contributes to adipocyte differentiation.

## Discussion

In this study, we demonstrated that heterozygous deletion of Cnot4 in mice attenuates HFD-induced obesity with decreased adipose tissue mass and hepatic fat depots compared with WT mice. Cnot4 Het MEFs significantly impaired differentiation into adipocytes. Cnot4 positively regulates PPARγ transcriptional activity, promoting the expression of adipogenic genes necessary for adipocyte differentiation.

In mouse spermatogenesis, CNOT4 was shown to act as an mRNA adaptor for CCR4-NOT complex during male germ cell meiosis by binding to subset of mRNAs and recruiting them to CCR4-NOT [24]. In obese conditions, knockout mice of CNOT7, a deadenylase subunit in the complex, were resistant to HFD-induced obesity in mice due to increased energy expenditure driven by up-regulated mRNA level of uncoupling protein 1 (Ucp1) in WATs [27]. Deletion of CNOT6L deadenylase upregulates expression of *Gdf15* and *Fgf21*, leading to appetite suppression via stimulation of the hindbrain and increased energy expenditure and lipid consumption, respectively [26]. However, there was no difference in the mRNA levels of those genes detected between WT and Cnot4 Het mice (Fig 3E). Thus, Cnot4 is unlikely to function as a recruiter of mRNAs for CCR4-NOT complex to deadenylate in the context of HFD-induced obesity.

Cnot4 Het mice exhibited the reduced size of BATs under basal conditions (normal diet feeding), implicating that Cnot4 has distinct roles in differentiation into brown adipocytes in BATs from WAT adipocyte differentiation. BATs contain more abundant mitochondria than WATs and thereby play an essential role in thermogenesis. Mitophagy is one of the key mechanisms for maintaining mitochondrial quality by degrading damaged mitochondria. It was recently reported that mitochondrial damage stalls translation of the nuclear-encoded mitochondrial gene *C-I30* mRNA on mitochondrial outer membrane, recruiting quality control factors Pelo, ABCE1 and CNOT4 to the stalled ribosome [33]. CNOT4-mediated ubiquitination of ABCE1 generates poly-ubiquitin signals, attracting autophagy receptors to initiate mitophagy. Thus, it may be speculated that small BATs in Cnot4 Het mice may be attributed to impaired mitochondria quality control and subsequent insufficiency of brown adipocyte differentiation. Further studies are needed to dissect the specific roles of CNOT4 in mitochondrial regulation in brown adipocytes.

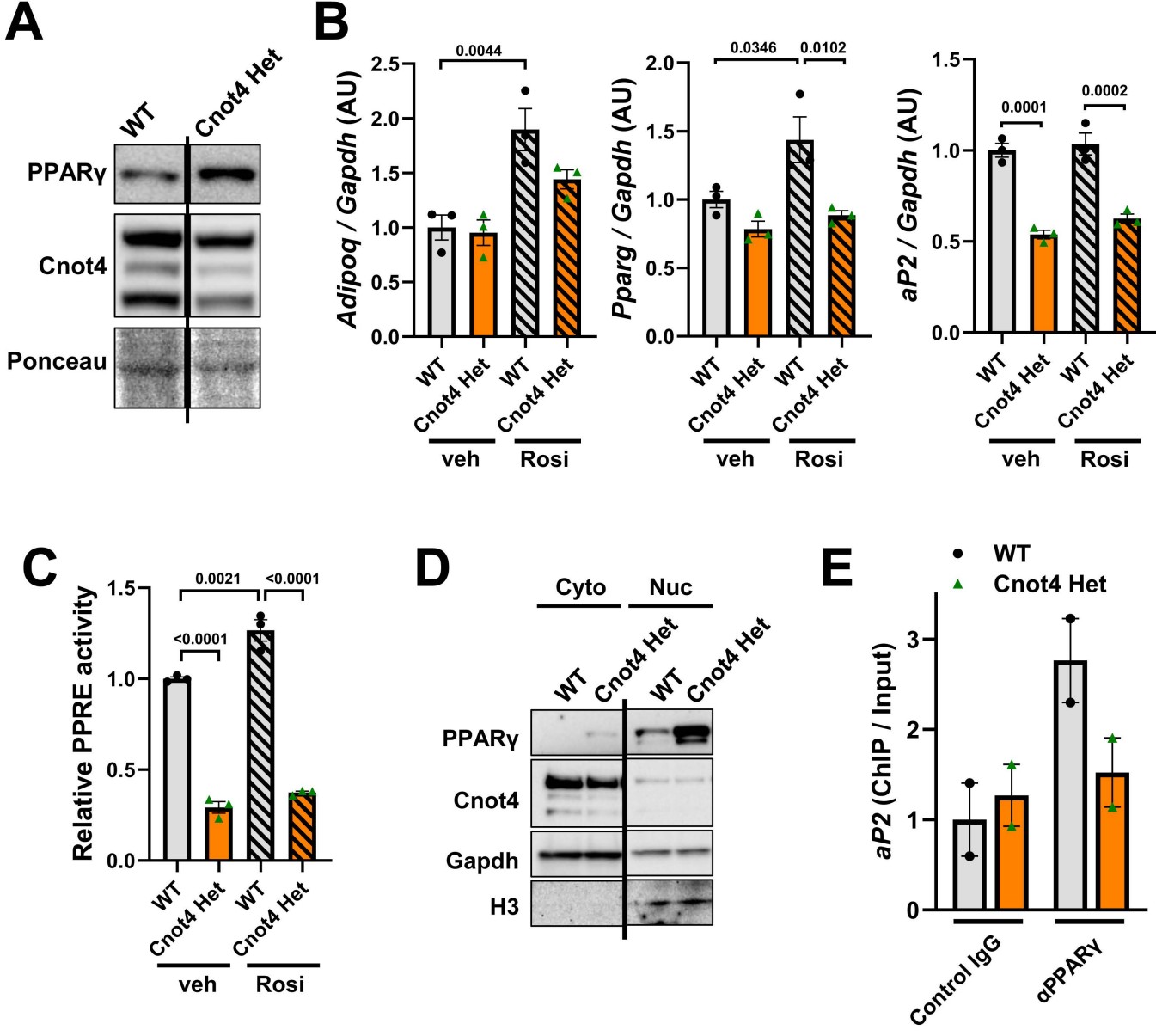

**Fig 5. Cnot4 haploinsufficiency impairs rosiglitazone-induced transcriptional activity of PPAR γ.** (A) Protein expression of PPARγ in MEFs. (B) mRNA levels in MEFs at 6 hours after 1 μM rosiglitazone treatment (n = 3). (C) Reporter assay using the luciferase reporter plasmid harboring PPARγ response element (PPRE). The reporter plasmid was transfected into MEFs at 24 hours before treatment of 1 μM rosiglitazone, and cells were harvested 6 hours after rosiglitazone treatment (n = 3). (D) Western blot of PPARγ in cytoplasm (Cyto) and nuclear (Nuc) fraction of MEFs. (E) ChIP-qPCR for PPARγ-bound *aP2* promoter regions in MEFs (n = 2). All values are means ± SEM. One-way ANOVA with Sidak's multiple-comparisons test (B and C). Numbers above square brackets show significant *P*-values.

The present study shows that Cnot4 Het mice appear resistant to obesity under HFD feeding. In enlargement of adipose tissues during obesity, Cnot4 heterozygosity may primarily suppresses hyperplasia, which has minimal metabolic impact compared to hypertrophy [1]. This idea would be supported by two observations; homozygous Cnot4 knockout causes embryonic lethality, implicating the essential roles of CNOT4 in general cellular development, and Cnot4 heterozygosity down-regulates the expression of adipogenic genes in MEFs while no significant effects on WATs in vivo,

suggesting the importance of CNOT4 in immature preadipocytes but not in mature adipocytes. As for discrepancy of in vitro and in vivo results, when mRNA expression levels of adipogenic genes in MEFs (Fig 4E) were further normalized with the lipid contents quantified with Oil Red O absorbance (Fig 4D), the expression levels of adipogenic genes were comparable between Cnot4 Het MEFs and WT MEFs (not shown). Thus, decreased mRNA expression of adipogenic genes in Cnot4 Het MEFs (Fig 4E) seems to reflect impaired differentiation into adipocytes, and Cnot4 heterozygosity no longer influences adipogenic gene expression in mature adipocytes.

Comparable glucose intolerance and serum lipid levels in Cnot4 Het mice and WT mice but resistance to obesity in Cnot4 Het mice is intriguing, whereas these phenotypes are partially similar to PPARγ heterozygous knockout (*PPARγ^+/−*) mice, which exhibited resistance to HFD-induced obesity and increased sensitivity to insulin but little or no improvement in glucose intolerance [34]. Although plasma triglyceride levels have not been measured in *PPARγ^+/−* mice, the phenotypic similarities may further support our proposal of CNOT4-mediated regulation of PPARγ activity. Mechanistically, one can speculate that slight but significant increase in mRNA levels of *Pgc1a* and *Igfbp1* in the livers of Cnot4 Het mice might be related the phenotypes (Fig 3E). PGC-1α is a master regulator of mitochondrial biogenesis and function [35], and upregulation of PGC-1α expression leads to enhanced lipid oxidation and energy expenditure, which may potentially counterbalance the high serum lipid levels redistributed through reduced lipid storage capacity in the smaller fat tissues in Cnot4 Het mice. IGFBP1 modulates insulin-like growth factor 1 (IGF-1)/insulin signaling by binding to IGF-1 and reducing its bioavailability, thereby down-modulating insulin sensitivity [36]. Elevated hepatic IGFBP1 may have neutralized IGF-1 in the livers of Cnot4 Het mice, leading to hepatic insulin resistance and inhibition of glucose uptake, and eventually augmenting glucose intolerance in Cnot4 Het mice. Further studies are needed to clarify the precise roles of Cnot4 in metabolic homeostasis.

PPARγ was identified as one of the Cnot4 targets in preadipocytes in this study. Nuclear import of PPARγ was not impaired, but the accessibility of PPARγ to *aP2* promoter was decreased in Cnot4 Het MEFs compared with WT MEFs, implicating that Cnot4 is crucial for recruitment of PPARγ to its target locus. On the other hand, our biochemical analysis did not detect the direct interaction of Cnot4 and PPARγ (not shown), and thus we assume that Cnot4 indirectly contributes to recruitment of PPARγ to target gene locus. Yeast CNOT4 (Not4) promotes ubiquitination of Jhd2, a demethylase for histone H3 lysine 4 methylation, and subsequent proteasomal degradation [19,37]. Not4 also regulates ubiquitination of Rpb1, the largest subunit of RNA polymerase II (RNAPII), halting transcriptional elongation and promoting RNAPII turnover during genotoxic stress [20]. Additionally, Not4 modulates the protein levels of polymerase-associated factor 1 (PAF1), a key RNAPII interactor [38]. Although those results are obtained for yeast Not4, it can be speculated that similar mechanisms are conserved for mammalian CNOT4-mediated regulation of PPARγ transcriptional activity.

## Conclusion

Cnot4 promotes adipocyte differentiation partly through PPARγ, and it likely contributes to hyperplasia in enlargement of adipose tissues during obesity. This seems to explain the reason that Cnot4 Het mice under HFD feeding are resistant to obesity but do not improve glucose intolerance and serum lipid levels. Further mechanistic insights into Cnot4-mediated activation of PPARγ may lead to a better understanding of PPARγ biology and adipocyte differentiation.

## Supporting information

**S1 raw images. A, Uncropped image of gel in Fig 1B.** Yellow rectangle area is shown in the Figure. B, Uncropped images of the blot in Fig 5A. Black rectangle area is shown in the Figure. C, Uncropped images of the blot in Fig 5D. Black rectangle area is shown in the Figure.
(TIF)

**S1 Table. Primer sequences for genotyping.**
(DOCX)

**S2 Table. Primer sequences for qPCR.**

(DOCX)

AcknowledgmentssWe thank all members of our laboratories for technical assistance and helpful discussions.

## Author contributions

**Conceptualization:** Tomokazu Yamaguchi, Keiji Kuba.

**Data curation:** Tomokazu Yamaguchi, Keiji Kuba.

**Formal analysis:** Tomokazu Yamaguchi, Keiji Kuba.

**Funding acquisition:** Tomokazu Yamaguchi, Keiji Kuba.

**Investigation:** Tomokazu Yamaguchi, Midori Hoshizaki, Keiji Kuba.

**Methodology:** Yumiko Imai, Tadashi Yamamoto.

**Resources:** Tadashi Yamamoto.

**Supervision:** Keiji Kuba.

**Writing – original draft:** Tomokazu Yamaguchi, Keiji Kuba.

**Writing – review & editing:** Tomokazu Yamaguchi, Keiji Kuba.

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
