## [Decision Letter · Decision Letter 0]

29 Dec 2024

PONE-D-24-57032Cnot4 heterozygosity attenuates high fat diet-induced obesity in mice and impairs PPARγ-mediated adipocyte differentiation.PLOS ONE

Dear Dr. Kuba,

Thank you for submitting your manuscript to PLOS ONE. After careful consideration, we feel that it has merit but does not fully meet PLOS ONE’s publication criteria as it currently stands. Therefore, we invite you to submit a revised version of the manuscript that addresses the points raised during the review process.

We look forward to receiving your revised manuscript.

Kind regards,

Palash Mandal

Academic Editor

PLOS ONE

Journal Requirements:

Additional Editor Comments :

Dear Authors,

Thank you for submitting your manuscript to PLOS ONE. After careful consideration, we feel that it has merit but does not fully meet PLOS ONE publication criteria as it currently stands. The shortcomings of this paper needs to be worked out before it can be considered for publication. Therefore, we invite you to resubmit a revised version of the manuscript that addresses the points raised during the review process.

For your guidance, the reviewers' comments are included below.

Thank you for giving us the opportunity to consider your work.

Specific concerns expressed during peer review were:

Reviewers' comments:

Reviewer's Responses to Questions

**Comments to the Author**

1. Is the manuscript technically sound, and do the data support the conclusions?

Reviewer #1: Yes

Reviewer #2: Partly

2. Has the statistical analysis been performed appropriately and rigorously? 

Reviewer #1: Yes

Reviewer #2: Yes

3. Have the authors made all data underlying the findings in their manuscript fully available?

Reviewer #1: Yes

Reviewer #2: Yes

4. Is the manuscript presented in an intelligible fashion and written in standard English?

Reviewer #1: Yes

Reviewer #2: Yes

5. Review Comments to the Author

Reviewer #1: This manuscript investigates the role of CNOT4, an E3 ubiquitin ligase, in adipocyte differentiation and obesity, using Cnot4 heterozygous knockout mice. The study finds that Cnot4 heterozygosity attenuates high-fat diet-induced obesity by impairing adipocyte differentiation, and that this effect is mediated by the reduced transcriptional activity of PPARγ. While the study presents some interesting findings, several concerns arise regarding its originality, methodological rigor, and interpretation of results.

While the manuscript claims that the role of CNOT4 in obesity is unexplored, the introduction cites studies on other components of the CCR4-NOT complex (CNOT3, CNOT6L, and CNOT7) that are involved in obesity and energy metabolism, which raises questions about the novelty of exploring another member of this complex. Although this is by itself not a huge issue.

The study suggests that up-regulating CNOT4 might benefit lipodystrophy patients, which seems counterintuitive given their finding that CNOT4 promotes adipocyte differentiation. If CNOT4 is necessary for adipocyte differentiation, then, it is not clear how its upregulation would treat lipodystrophy, which is characterized by loss of adipose tissue. The authors need to clarify this point.

The study proposes that understanding CNOT4's mechanism in activating PPARγ could lead to new drugs for obesity, diabetes, and cardiovascular disease. However, the study does not show a direct interaction between CNOT4 and PPARγ. The assertion that CNOT4’s role in PPARγ regulation could lead to new therapies requires stronger evidence than what the manuscript currently provides.

The study concludes that Cnot4 heterozygosity leads to reduced adipocyte differentiation using in vitro MEF experiments, but the study shows that Cnot4 Het mice did not show significant change in the expression of metabolic genes in WATs and livers. This needs to be explained.

The authors claim that Cnot4 heterozygous deletion suppressed adipocyte differentiation partly through reduced transcriptional activity of PPARγ. However, Cnot4 Het mice did not show improved dyslipidemia or glucose intolerance. If PPARγ is crucial for adipogenesis, it is not clear why dyslipidemia or glucose intolerance were not affected despite changes in PPARγ activity.

The study does not investigate the specific mechanisms behind how CNOT4 regulates PPARγ transcriptional activity. The study suggests a potential role of epigenetic modification or RNAPII regulation, however, these suggestions remain purely speculative.

The methods section lacks detail regarding the specific procedures for measuring mRNA expression, protein expression and luciferase activity. The manuscript mentions that 'qRT-PCR analysis was conducted as previously described' citing another paper, but this doesn’t allow the reader to properly assess the rigor of those methods.

The figure legends include some statistical tests. However, the data provided in the figures are sometimes not sufficient to justify the statistical tests they claim to have used.

There is an inconsistency in the data presented. For example, Figure 2A shows mRNA expression of Cnot4 in mouse livers for WT and Cnot4 Het1920, but that data is not used anywhere else in the results.

Figure 1: The figure legend describes the knockout strategy and shows genotyping results, but it doesn't include the genotypes of the parents of the embryos. This omission makes it difficult to assess whether the correct genotypes were obtained for the different crosses.

Figure 4: The representative images of adipocytes stained with Oil Red O are not clear. Although the areas of Oil Red O staining are quantified in a bar graph, the image clarity makes it difficult to ascertain the level of lipid accumulation in the cells.

Overall Recommendation: The study presents some interesting observations regarding CNOT4's role in adipocyte differentiation and obesity. However, there are significant concerns about the originality, the methodological rigor, the interpretation of results, and the statistical analysis. The manuscript requires major revisions as outlined above.

Reviewer #2: This study by Yamaguchi et al. investigates the role of CNOT4 in adipocyte differentiation and its impact on diet-induced obesity using a KO mouse model. The authors provide evidence that the KO mice exhibit resistance to diet-induced obesity, attributed to suppressed adipocyte hyperplasia by regulating PPAR� transcriptional activity. These findings suggest altered adipose tissue functionality. However, I have several concerns that require further clarification and discussion.

1. The authors do not address the fate of dietary lipids in the KO mice, which is crucial given their resistance to diet-induced obesity and smaller adipocyte size. If adipocyte differentiation and lipid storage are impaired, it is important to discuss the potential lipid redistribution and its implications for systemic metabolic health. The data indicate that ectopic lipid accumulation in the liver is absent and circulating lipid levels remain unaltered in the KO mice. These observations suggest that lipid oxidation or energy expenditure might be enhanced. Further investigation or at least a discussion of these possibilities is needed.

2. The quantification of adipocyte size should be presented more comprehensively (e.g., frequency histograms or size range percentages) to better illustrate the observed differences. Additionally, the authors should specify how many fields of view per slide and how many mice were analyzed. The same level of detail is necessary for Figure 4C.

3. In the second paragraph of the Discussion, the authors mention that there were no differences in the mRNA levels of Ucp1, Gdf15, and Fgf21 in Cnot4 Het mice (lines 171–172). However, I could not find any corresponding data in the manuscript. If these results have been published elsewhere, the authors should provide an appropriate citation. If the data are unpublished, they should explicitly state that it is an unpublished observation or “data not shown”.

4. The last paragraph of Discussion part, lines 207–209, does not make sense. It implies a causal relationship between the lack of improvement in glucose and lipid levels and the ineffectiveness of CNOT4 suppression for obesity treatment.

5. Please specify the name and version of the statistical software used for data analysis.

6. There are some typos in the manuscript. Line 188, the authors probably meant “down-regulate”, and line 422 (E) probably means weight of “hearts”.

6. PLOS authors have the option to publish the peer review history of their article (what does this mean? ). If published, this will include your full peer review and any attached files.

**Do you want your identity to be public for this peer review?** For information about this choice, including consent withdrawal, please see our Privacy Policy .

Reviewer #1: No

Reviewer #2: No

---

## [Author Response · Author response to Decision Letter 1]

2 Mar 2025

Point by point reply to Reviewer

ID: PONE-D-24-57032

Reviewer #1:

- This manuscript investigates the role of CNOT4, an E3 ubiquitin ligase, in adipocyte differentiation and obesity, using Cnot4 heterozygous knockout mice. The study finds that Cnot4 heterozygosity attenuates high-fat diet-induced obesity by impairing adipocyte differentiation, and that this effect is mediated by the reduced transcriptional activity of PPARγ. While the study presents some interesting findings, several concerns arise regarding its originality, methodological rigor, and interpretation of results.

1. While the manuscript claims that the role of CNOT4 in obesity is unexplored, the introduction cites studies on other components of the CCR4-NOT complex (CNOT3, CNOT6L, and CNOT7) that are involved in obesity and energy metabolism, which raises questions about the novelty of exploring another member of this complex. Although this is by itself not a huge issue.

[Reply] First of all, we would like to thank you for reviewing our manuscript and providing valuable comments. In yeast, CNOT4 functions as a component of the CCR4-NOT complex by tightly binding to CNOT1, the central scaffold of the complex. However, this interaction of CNOT4 and CNOT1 is absent in mammals, as mammalian CNOT4 lacks the domain to bind CNOT1. Consequently, the role of CNOT4 as a part of the CCR4-NOT complex has been controversial in mammals, and thus its physiological function has remained elusive. While CNOT3, CNOT6L and CNOT7 have been shown to contribute to energy metabolism by mediating mRNA deadenylation, CNOT4 has only been reported to function as a ubiquitin ligase or scaffold protein, with no evidence to suggest its involvement in deadenylation when we start this study. In fact, our pull-down experiments failed to see a protein interaction between CNOT4 and the CCR4-NOT components (not shown). To address if target mRNAs of CNOT4 overlap with those of CCR4-NOT components, we conducted a new experiment to examine whether haploinsufficiency of Cnot4 affects the expression of target mRNAs of Cnot3 and Cnot6l in previous reports. As a result, heterozygous deletion of Cnot4 modestly upregulated expression of two target mRNAs of Cnot3 among the 6 genes tested (new Fig 3E). Thus, Cnot4 is likely to regulate gene expression through the mechanisms distinct from the known mechanism of CCR4-NOT-mediated deadenylation and subsequent downregulation of gene expression, and thus we believe that this finding provides new insights into the unique function of CNOT4 in mammals. We added new data of Fig 3E and describe those in the results section (page 12; lines 252-261).

2. The study suggests that up-regulating CNOT4 might benefit lipodystrophy patients, which seems counterintuitive given their finding that CNOT4 promotes adipocyte differentiation. If CNOT4 is necessary for adipocyte differentiation, then, it is not clear how its upregulation would treat lipodystrophy, which is characterized by loss of adipose tissue. The authors need to clarify this point.

[Reply] Lipodystrophy is a medical condition characterized by abnormal loss of adipose tissue. If overexpression of CNOT4 gene in preadipocytes promotes its differentiation into mature adipocytes, it may help to improve increase fat mass in lipodystrophy patients. However, as you pointed out, we did not examine the effects of CNOT4 overexpression on adipocyte differentiation in this study. We would like to clarify this point in the future studies. Accordingly, we removed the description on potential lipodystrophy treatment from the Discussion section (page 18).

3. The study proposes that understanding CNOT4's mechanism in activating PPARγ could lead to new drugs for obesity, diabetes, and cardiovascular disease. However, the study does not show a direct interaction between CNOT4 and PPARγ. The assertion that CNOT4’s role in PPARγ regulation could lead to new therapies requires stronger evidence than what the manuscript currently provides.

[Reply] In this study, we found that CNOT4 positively regulates the transcriptional regulation of PPARγ. However, as you noted, we did not clarify the details of molecular mechanism. Further studies are needed to solve these issues. Thus, we have toned down our conclusion (page 19; line 425-430).

4. The study concludes that Cnot4 heterozygosity leads to reduced adipocyte differentiation using in vitro MEF experiments, but the study shows that Cnot4 Het mice did not show significant change in the expression of metabolic genes in WATs and livers. This needs to be explained.

[Reply] We appreciate the important comments. mRNA expression of adipogenic genes were decreased in Cnot4 Het MEFs compared with WT MEFs (Fig 4E). On the other hand, when mRNA levels were further normalized with lipid amounts quantified with Oil Red O absorbance (Fig 4D), the expression levels were comparable between WT MEFs and Cnot4 Het MEFs (data not shown). The results suggest that mRNA expression of adipogenic genes (Fig 4E) seems to reflect Cnot4’s roles in adipocyte differentiation and also implicate that heterozygosity of Cnot4 does not affect adipogenic gene expression in mature adipocytes. Accordingly, we modified the text in the Discussion section to describe that Cnot4 heterozygosity may primarily suppress hyperplasia with minimal effects on the matured adipocytes (page 17; lines 386-391).

5. The authors claim that Cnot4 heterozygous deletion suppressed adipocyte differentiation partly through reduced transcriptional activity of PPARγ. However, Cnot4 Het mice did not show improved dyslipidemia or glucose intolerance. If PPARγ is crucial for adipogenesis, it is not clear why dyslipidemia or glucose intolerance were not affected despite changes in PPARγ activity.

[Reply] Thank you for your valuable comment. No improvement in dyslipidemia or glucose intolerance but resistance to obesity are intriguing phenotypes of Cnot4 Het mice. On the other hand, it should be noted that these phenotypes are partially similar to PPARγ heterozygous knockout mice, which exhibited resistance to HFD-induced obesity and increased sensitivity to insulin but little or no improvement in glucose intolerance (Kubota N, Mol Cell 1999). Thus, the phenotypic similarities may further support our proposal of CNOT4-mediated regulation of PPARγ activity. Mechanistically, our new data revealed that mRNA expression of Igfbp1 is upregulated in the livers of Cnot4 Het mice (new Fig 3E). IGFBP1 modulates insulin-like growth factor 1 (IGF-1)/insulin signaling by binding to IGF-1 and reducing its bioavailability, thereby down-modulating insulin sensitivity. Elevated hepatic IGFBP1 may have neutralized IGF-1 in the livers of Cnot4 Het mice, leading to hepatic insulin resistance and inhibition of glucose uptake, and eventually augmenting glucose intolerance in Cnot4 Het mice. Nevertheless, further studies are needed to clarify the precise roles of Cnot4 in metabolic homeostasis. We described the new data (new Fig 3E) in the results section (page 12; lines 252-261) and included a new literature and our assumption in the discussion section (page 17; line 393 - page 18; line 409).

6. The study does not investigate the specific mechanisms behind how CNOT4 regulates PPARγ transcriptional activity. The study suggests a potential role of epigenetic modification or RNAPII regulation, however, these suggestions remain purely speculative.

[Reply] To address how CNOT4 regulates PPARγ transcriptional activity, we first conducted new experiments of the subcellular fractionation of PPARγ in MEFs to determine whether heterozygous knockout of Cnot4 suppress the nuclear translocation of PPARγ. As a result, we observed sufficient nuclear translocation of PPARγ in Cnot4 Het MEFs (new Fig. 5D). Next, we newly conducted chromatin immunoprecipitation (ChIP) assay and then found that heterozygosity of Cnot4 decreased the binding of PPARγ protein to the promoter regions of aP2 gene locus in MEFs (new Fig. 5E). On the other hand, we did not detect the direct interaction of Cnot4 and PPARγ by immunoprecipitation and Western Blot (Data not shown). Thus, Cnot4 is involved in the recruitment of PPARγ to its target gene locus possibly through epigenetic modifications and/or Pol II regulation. Further studies are needed to clarify the detailed mechanisms of this regulation though. Accordingly, we revised the text in the Results section (page 15; lines 328-336) and in the Discussion section (page 18; lines 411-416).

7. The methods section lacks detail regarding the specific procedures for measuring mRNA expression, protein expression and luciferase activity. The manuscript mentions that 'qRT-PCR analysis was conducted as previously described' citing another paper, but this doesn’t allow the reader to properly assess the rigor of those methods.

[Reply] We provided additional details on the procedures for measuring mRNA expression (page 7; lines 136-138), protein expression (page 7; line 147 - page 8; line 157), and luciferase activity (page 8; lines 160-168).

8. The figure legends include some statistical tests. However, the data provided in the figures are sometimes not sufficient to justify the statistical tests they claim to have used.

[Reply] We apologize for the error of description on statistical analyses. We corrected the statistical test in Figure 2 (page 11; line 235).

9. There is an inconsistency in the data presented. For example, Figure 2A shows mRNA expression of Cnot4 in mouse livers for WT and Cnot4 Het1920, but that data is not used anywhere else in the results.

[Reply] Apology for confusing data citation. We corrected the text to explain the Figure 2A (page 10; lines 213-214).

10. Figure 1: The figure legend describes the knockout strategy and shows genotyping results, but it doesn't include the genotypes of the parents of the embryos. This omission makes it difficult to assess whether the correct genotypes were obtained for the different crosses.

[Reply] As pointed out, we revised the text to add genotypes of the parents of the embryos in the legend of Figure 1 (page 10; lines 202-206).

11. Figure 4: The representative images of adipocytes stained with Oil Red O are not clear. Although the areas of Oil Red O staining are quantified in a bar graph, the image clarity makes it difficult to ascertain the level of lipid accumulation in the cells.

[Reply] We apologize for unclear images of Oil Red O staining. We included the images with higher magnification and better quality in Figure 4C.

Overall Recommendation: The study presents some interesting observations regarding CNOT4's role in adipocyte differentiation and obesity. However, there are significant concerns about the originality, the methodological rigor, the interpretation of results, and the statistical analysis. The manuscript requires major revisions as outlined above.

[Reply] Thank you very much for reviewing and evaluating our manuscript. According to your comments, we conducted the experiments including qPCR, western blot, and ChIP assay, and revised the text to address all comments. I sincerely appreciate your valuable comments, as it has significantly improved our paper.

Reviewer #2:

- This study by Yamaguchi et al. investigates the role of CNOT4 in adipocyte differentiation and its impact on diet-induced obesity using a KO mouse model. The authors provide evidence that the KO mice exhibit resistance to diet-induced obesity, attributed to suppressed adipocyte hyperplasia by regulating PPARg transcriptional activity. These findings suggest altered adipose tissue functionality. However, I have several concerns that require further clarification and discussion.

1. The authors do not address the fate of dietary lipids in the KO mice, which is crucial given their resistance to diet-induced obesity and smaller adipocyte size. If adipocyte differentiation and lipid storage are impaired, it is important to discuss the potential lipid redistribution and its implications for systemic metabolic health. The data indicate that ectopic lipid accumulation in the liver is absent and circulating lipid levels remain unaltered in the KO mice. These observations suggest that lipid oxidation or energy expenditure might be enhanced. Further investigation or at least a discussion of these possibilities is needed.

[Reply] We would like to thank you for reviewing our manuscript and pointing out important aspects. We agree that it is important to address the potential redistribution of dietary lipids in Cnot4 Het mice. In this revision, we investigated the expression of mRNAs encoding energy metabolism-related genes in the livers, which had been previously reported to be regulated by CCR4-NOT components and thus are thought to be potentially related to Cnot4. As a result, among the genes examined, Pgc1a was found to be modest but significantly up-regulated in the livers of Cnot4 Het mice compared with WT mice (new Fig. 3E). Since PGC-1α is a master regulator of mitochondrial biogenesis, we speculate that upregulation of PGC-1α expression may lead to enhanced lipid oxidation and energy expenditure, thereby down-modulating the lipids accumulated in the livers of Cnot4 Het mice. We described the new data in the results section (page 12; lines 252-261) and our assumption in the discussion section (page 17; line 393 - page 18; line 409).

2. The quantification of adipocyte size should be presented more comprehensively (e.g., frequency histograms or size range percentages) to better illustrate the observed differences. Additionally, the authors should specify how many fields of view per slide and how many mice were analyzed. The same level of detail is necessary for Figure 4C.

[Reply] As recommended, we comprehensively quantified the size of adipocytes in WATs of mice (new Fig 2J, page 11; lines 244-245). In addition, we revised the text to mention how many fields (page 6; lines 109-113) and mice (page 11; lines 234-235) were used to quantify adipocyte size. We also revised the text to provide additional details on the procedures for Oil Red O area quantification in Figure 4C, as pointed out (page 6; line 126 - page 7; line 130).

3. In the second paragraph of the Discussion, the authors mention that there were no differences in the mRNA levels of Ucp1, Gdf15, and Fgf21 in Cnot4 Het mice (lines 171–172). However, I could not find any corresponding data in the manuscript. If these results have been published elsewhere, the authors should provide an appropriate citation. If the data are unpublished, they should explicitly state that it is an unpublished observation or “data not shown”.

[Reply] As recommended, we have added the data on Gdf15 and Fgf21 mRNA expression in the liver. (new Fig 3E, page 12; lines 252 - 258).

4. The last paragraph of Discussion part, lines 207–209, does not make sense. It implies a causal relationship between the lack of improvement in glucose and lipid levels and the ineffectiveness of CNOT4 suppression for obesity treatment.

[Reply] As pointed out, we have not observed improvement in glucose and lipid levels in Cnot4 Het mice, which may be due to altered metabolic gene expression in the livers. Further studies, such as to determine whether adipose tissue-specific suppression of Cnot4 is beneficial for obesity treatment would be needed. We withdrew our claims and mentioned the necessity of further studies in the conclusion (page 19; lines 425 - 430).

5. Please specify the name and version of the statistical software used for data analysis.

[Reply] As recommended, we added the name and version of the statistical software in Materials and methods part (page 9; line 184-185).

6. There are some typos in the manuscript. Line 188, the authors probably meant “down-regulate”, and line 422 (E) probably means weight of “hearts”.

[Reply] We apologize for several typing errors. We corrected the errors (page 11; line 229).

---

## [Decision Letter · Decision Letter 1]

24 Apr 2025

Cnot4 heterozygosity attenuates high fat diet-induced obesity in mice and impairs PPARγ-mediated adipocyte differentiation.

PONE-D-24-57032R1

Dear Dr. Kuba,

We’re pleased to inform you that your manuscript has been judged scientifically suitable for publication and will be formally accepted for publication once it meets all outstanding technical requirements.

Kind regards,

Nobuyuki Takahashi, Ph.D.

Academic Editor

PLOS ONE

Additional Editor Comments (optional):

Reviewers' comments:

Reviewer's Responses to Questions

**Comments to the Author**

1. If the authors have adequately addressed your comments raised in a previous round of review and you feel that this manuscript is now acceptable for publication, you may indicate that here to bypass the “Comments to the Author” section, enter your conflict of interest statement in the “Confidential to Editor” section, and submit your "Accept" recommendation.

Reviewer #1: All comments have been addressed

Reviewer #2: All comments have been addressed

2. Is the manuscript technically sound, and do the data support the conclusions?

Reviewer #1: Yes

Reviewer #2: Yes

3. Has the statistical analysis been performed appropriately and rigorously? 

Reviewer #1: Yes

Reviewer #2: Yes

4. Have the authors made all data underlying the findings in their manuscript fully available?

Reviewer #1: Yes

Reviewer #2: Yes

5. Is the manuscript presented in an intelligible fashion and written in standard English?

Reviewer #1: Yes

Reviewer #2: Yes

6. Review Comments to the Author

Reviewer #1: Some minor remaining concerns:

The manuscript continues to suggest that CNOT4 regulates PPARγ activity through recruitment to chromatin, but no direct or indirect co-factor interaction is shown. The absence of direct interaction is acknowledged, but the discussion still speculates heavily about epigenetic or polymerase-associated mechanisms based on yeast orthologs. This should be clearly labeled as a hypothesis.

The authors wisely toned down their claims, but occasional phrases still hint at clinical relevance (e.g., implications for obesity treatment) that go beyond the evidence.

While addressed, the manuscript would benefit from a clearer, less technical explanation of why Cnot4 Het affects differentiation in MEFs but not mature adipocyte gene expression in vivo. This distinction is central to interpreting the model.

Reviewer #2: (No Response)

7. PLOS authors have the option to publish the peer review history of their article (what does this mean? ). If published, this will include your full peer review and any attached files.

**Do you want your identity to be public for this peer review?** For information about this choice, including consent withdrawal, please see our Privacy Policy .

Reviewer #1: No

Reviewer #2: No

---

## [Editor Report · Acceptance letter]

PONE-D-24-57032R1

PLOS ONE

Dear Dr. Kuba,

I'm pleased to inform you that your manuscript has been deemed suitable for publication in PLOS ONE. Congratulations! Your manuscript is now being handed over to our production team.

Kind regards,

on behalf of

Dr. Nobuyuki Takahashi

Academic Editor

PLOS ONE